# RAB20 Promotes Proliferation via G2/M Phase through the Chk1/cdc25c/cdc2-cyclinB1 Pathway in Penile Squamous Cell Carcinoma

**DOI:** 10.3390/cancers14051106

**Published:** 2022-02-22

**Authors:** Xingliang Tan, Gangjun Yuan, Yanjun Wang, Yuantao Zou, Sihao Luo, Hui Han, Zike Qin, Zhuowei Liu, Fangjian Zhou, Yanling Liu, Kai Yao

**Affiliations:** 1Department of Urology, Sun Yat-sen University Cancer Center, Guangzhou 510060, China; tanxl1@sysucc.org.cn (X.T.); wangyj@sysucc.org.cn (Y.W.); zouyt1@sysucc.org.cn (Y.Z.); luosh@sysucc.org.cn (S.L.); hanhui@sysucc.org.cn (H.H.); qinzk@sysucc.org.cn (Z.Q.); liuzhw@sysucc.org.cn (Z.L.); zhoufj@sysucc.org.cn (F.Z.); 2State Key Laboratory of Oncology in Southern China, Guangzhou 510060, China; 3Collaborative Innovation Center of Cancer Medicine, Guangzhou 510060, China; 4Department of Urology Oncological Surgery, Chongqing University Cancer Hospital, Chongqing 400030, China; yuangj@cqu.edu.cn; 5Chongqing Key Laboratory of Translational Research for Cancer Metastasis and Individualized Treatment, Chongqing University Cancer Hospital, Chongqing 400030, China; 6Department of Operating Room, Sun Yat-sen University Cancer Center, Guangzhou 510060, China

**Keywords:** RAB20, penile squamous cell carcinoma, cell proliferation, cell cycle, prognostic biomarker

## Abstract

**Simple Summary:**

There is no available biomarker that could be used to predict the prognosis or progression of penile squamous cell carcinoma (PSCC) currently. Here, we conducted comprehensive genomic sequencing on eight pairs of PSCC tissues and found that RAB20 was upregulated in tumors, especially in metastatic lymph nodes. We then detected the expression of RAB20 in a large cohort of 259 PSCC cases using IHC assay and analyzed the association between the RAB20 expression and clinical features. We found that high RAB20 expression predicted poor prognosis and advanced clinicopathological features. Further in vitro and in vivo tumorigenesis assays revealed that RAB20 regulates cell proliferation at the G2/M cell cycle phase through the Chk1/cdc25c/cdc2-cyclinB1 pathway. Our results indicated that RAB20 could be a promising prognostic biomarker of advanced PSCC with poor survival and could be a potential therapeutic target.

**Abstract:**

RAB20, a member of the RAS GTPase oncogene family, is overexpressed in several cancers with poor outcomes, promoting tumorigenesis and inducing genomic instability. Here, we performed comprehensive genomic sequencing on eight penile squamous cell carcinoma (PSCC) and normal tissue pairs and found that RAB20 was upregulated in tumors, especially in metastatic lymph nodes. RAB20 overexpression in tumors was further verified by qPCR, Western blotting, and immunohistochemistry of our newly established PSCC cell lines and paired tissues. The clinical significance of RAB20 was validated in 259 PSCC patients, the largest cohort to date, and high RAB20 expression positively correlated with the T, N, M status, extranodal extension, and clinical stage (all *p* < 0.01). RAB20 was an unfavorable independent prognostic indicator in the survival analysis (*p* = 0.011, HR = 2.090; 95% Cl: 1.183–4.692), and PSCC patients with high RAB20 expression experienced shorter 5-year cancer-specific survival times (*p* < 0.001). Furthermore, tumorigenesis assays demonstrated that RAB20 knockdown inhibited cell proliferation, migration, and colony formation in vitro and tumor growth in vivo. RAB20 depletion also induced PSCC cell cycle arrest at G2/M by increasing Chk1 expression and promoting cdc25c phosphorylation to reduce cdc2-cyclinB1 complex formation. Our study revealed an oncogenic role for RAB20 in promoting PSCC cell proliferation at the G2/M phase via the Chk1/cdc25c/cdc2-cyclinB1 pathway. Thus, RAB20 could be a promising prognostic biomarker of advanced PSCC with poor patient survival outcomes and could be a potential therapeutic target.

## 1. Background

Penile squamous cell carcinoma (PSCC), with a global incidence of 0.4–0.6/100,000, is a devastating genitourinary disease in males [1,2]. Compared with the histologic features and pathologic stages of the primary tumor, lymph node metastasis is the most unfavorable factor affecting long-term survival outcomes, and PSCC patients have a dismal 5-year cancer-specific survival rate (CSS) of approximately 29–59% [3,4,5,6]. Currently, researchers face significant challenges and difficulties in investigating the progression and invasion of PSCC owing to the tumor heterogeneity [4,7,8]. Although several genomic studies have revealed that EGFR amplification and CDKN2A mutations are frequent somatic alterations in metastatic PSCC, their clinical significance and expression profiles during tumor progression remain elusive owing to noncomprehensive transcriptome sequencing methods, small sample sizes, and unpaired PSCC tissues [8,9,10]. Overexpression of BIRC5 [11], IDO1 [12], and LAMC2 [13] was found to be associated with advanced disease and poor prognosis, enhancing PSCC cell proliferation and invasion, but the detailed mechanism of tumor progression remains unclear. Despite research progress, the lack of appropriate cell lines and large-scale clinical validation methods greatly hinders the exploration of biomarkers associated with the PSCC progression.

To identify potential prognostic markers of PSCC and intrinsic mechanisms, eight pairs of pN+ PSCC tissues (normal tissues, primary tumor, and metastatic lymph node) were subjected to whole-transcriptome comprehensive genomic profiling (CGP). We found that RAB20 was the critical oncogene overexpressed in PSCC tissues, especially in metastatic lymph nodes, and was associated with poor survival. However, the roles of RAB20 and its specific mechanisms in PSCC are still unknown.

RAB20 belongs to the Rab family of small GTPases, which plays a critical role in membrane trafficking in epithelial cells and has been found to be associated with the progression of several cancers [14]. Overexpression and amplification of RAB20 have been detected in pancreatic carcinoma, colorectal adenoma, and triple-negative breast cancer and are associated with high-risk clinicopathological stages and poor survival outcomes [15,16,17].

In this study, we investigated the expression pattern of RAB20 in PSCC tissues and demonstrated a correlation between RAB20 expression and clinicopathological features in 259 PSCC patients, the largest cohort reported to date. We identified that RAB20 is an independent prognostic indicator correlated with poor survival outcomes. Furthermore, by using newly established PSCC cell lines and animal models [18], we demonstrated that RAB20 promotes cell proliferation and tumor progression, inducing the G2/M phase cell cycle via the Chk1/cdc25c/cdc2-cyclinB1 pathway.

## 2. Materials and Methods

### 2.1. Patient Cohort, Samples, and Research Ethics

This study included a total of 259 patients who were diagnosed with pathologically confirmed PSCC at the Sun Yat-sen University Cancer Center (SYSUCC) between January 2000 and December 2019. For each patient, their clinical, pathological, and survival information was retrospectively reviewed according to the TNM Staging System for Penile Cancer (8th ed., 2017). For human tissue samples, 8 pairs of pN+ PSCC matched tissues were retrieved for CGP. A total of 99 fresh frozen samples, including 78 tumor tissues and 21 normal tissues (among them 15 were paired tissues), were collected for mRNA and protein extraction. Paraffin-embedded tumor sections from 259 PSCC patients were re-analyzed by two independent pathologists (CCB and LLL), and immunohistochemistry (IHC) staining was performed.

### 2.2. Target Gene Screening for the Progression of PSCC

Quality-approved paired samples from 8 pN+ PSCC patients (including adjacent normal (N), primary carcinoma (PCA), and metastatic lymph node tissues (LM) were subjected to next-generation CGP with Affymetrix Microarrays (Shanghai Genechem Co., Ltd., Shanghai, China) [19]. We defined differentially expressed genes as the absolute value of fold change (FC) ≥ 2 according to the Benjamini–Hochberg method [20]. Then, we screened for target genes that were consistently overexpressed in the LM and PCA groups, in particular those higher in the LM group (FC:LM ≥ PCA and PCA:N ≥ 2.5; *p* < 0.05, and FDR < 0.05). The mRNA expression of potential oncogenes was further verified in the Gene Expression Omnibus (GEO) database (GSE57955), and intersection genes were selected for subsequent studies [21].

### 2.3. Cell Lines, Culture Conditions, and Transfection Methods

Five PSCC cell lines, Penl1, Penl2, 149rca, 149rm, and 156lm, were established in our laboratory as previously reported [18]. The normal control human epidermis keratinocyte (HaCaT) cell line was purchased from the Type Culture Collection of the Chinese Academy of Sciences (Shanghai, China). All cell lines were cultured in Dulbecco’s modified Eagle’s (DMEM) medium with 10% fetal bovine serum (FBS, Gibco, Waltham, MA, USA). To knock down the expression of target genes, PSCC cell lines were transfected with negative control (NC) short hairpin RNA plasmids (shRNA), shRAB20 interference plasmids (GV248; hU6-MCS-Ubiquitin-EGFP-IRES-puromycin), RAB20 overexpression plasmids (RAB20-3FLAG-IRES2-EGFP), negative vector plasmids, and small interfering RNA (siRNA). The effective sequences were as follows: RAB20-sh2, ATCCTCACCTATGATGTGAAT; RAB20-sh3, AAGGAAGAGTGCAGTCCCAAT and shNC, TTCTCCGAACGTGTCACGT. The siRNA sequence GCAACAGTATTTCGGTATAAT was used to knock down the expression of CHK1.

### 2.4. Gene-Set Enrichment Analysis (GSEA)

Penl2 PSCC cells were transfected with shRNA (RAB20-sh3 and shNC) to knock down the expression of RAB20, and RNA sequencing (RNA-seq) gene expression analysis was performed. Then, GSEA was conducted on normalized RNA-seq data (RAB20sh3 vs. shNC) by GSEA tools version 4.1 (http://www.broadinstitute.org/gsea, accessed on 24 April 2021). We analyzed the subsets of the Molecular Signatures Database (C2 and C5) related to cell cycle and cell proliferation and calculated the normalized enrichment score [22] and the corresponding *p*-value of the false discovery rate (FDR).

### 2.5. Immunohistochemistry Assay

In brief, 4 µm paraffin-embedded tissue sections from 259 PSCC patients were deparaffinized in xylene and rehydrated with an alcohol gradient. Antigens were restored by incubating in citrate buffer, adjusting the pH to 6, and heating for 15 min. Nonspecific antigens were blocked with QuickBlock™ Blocking Buffer (Beyotime, Nantong, China) for 15 min. Afterward, sections were incubated with RAB20 antibody (Abcam, ab197209, 1:1000, Hangzhou, China) overnight at 4 °C, followed by a 2 h incubation with horseradish peroxidase (HRP)-labeled goat anti-rabbit secondary antibody (Beyotime). Finally, immunochemical staining was visualized by a peroxidase EnVision Detection Kit (Dako, Glostrup, Denmark). Gene expression was judged independently by two pathologists (CCB and LLL) blind to the sample identities. RAB20 proteins were mainly localized in the cytoplasm. Therefore, the expression level of RAB20 in tumor was based on the cytoplasm staining score, which was multiplied by the staining intensity (0 for no staining, 1 for weak staining, 2 for moderate staining, and 3 for strong staining) and staining area (1 for 1–10%, 2 for 11–40%, 3 for 41–70%, and 4 for 71% above). The cutoff value was calculated by X-Tile software (Version 3.6.1) by standard Monte Carlo simulation methods [23]. In our cohorts, the cutoff value of RAB20 IHC scores was 4 points. It indicated that 0–4 points were regarded as low RAB20 expression and 6–12 points were high RAB20 expression.

### 2.6. Western Blot (WB)

Proteins were extracted by RIPA lysis buffer (Beyotime) with 1% phosphatase and protease inhibitors and separated by 10% SDS-PAGE (EpiZyme, Cambridge, MA, USA). Then, we transferred the proteins onto PVDF membranes (Pierce Biotechnology, Waltham, MA, USA) and blocked the membranes with 5% milk at 37 °C for 1 h. The membranes were incubated with primary antibody overnight at 4 °C. A subsequent 2 h incubation in secondary antibody was performed, and the membranes were then exposed to ECL reagents (Abcam, Cambridge, UK). Antibodies and dilutions were as follows: RAB20 antibody (Abcam, ab197209, 1:1000); α-tubulin (CST, #2144, 1:1000); β-actin (CST, #3700, 1:1000); CDK2 (CST, #2546, 1:1000); CyclinE1 (CST, #20808, 1:1000); CyclinD1 (CST, #55506, 1:1000); CyclinB1 (CST, #12231, 1:1000); cdc2 (CST, #9116, 1:1000); Chk1 (CST, #2360, 1:1000); cdc25C (CST, #4688, 1:1000); Phospho-cdc25C (CST, #4901, 1:1000); p53 (CST, #2527, 1:1000); Phospho-p53 (CST, #9286, 1:1000); and p21Waf1/Cip1 (CST, #2947, 1:1000).

### 2.7. Quantitative Real-Time Polymerase Chain Reaction (qPCR) Assay

The extraction (HiPure Total RNA Plus Micro Kit, Magen, Yushu, China), reverse transcription (HiScript Q RT SuperMix Kit, Vazyme, Nanjing, China), and amplification (ChamQ SYBR qPCR Green Master Mix Kit, Vazyme) were performed according to the manufacturer’s directions. The relative mRNA expression levels of the target genes were calculated by the 2^(−∆∆Ct)^ method and normalized against the expression level of GAPDH. The primers used for the corresponding genes were as follows: GAPDH forward 5′-TGGTGAAGACGCCAGTGGA-3′ and reverse 5′-GCACCGTCAAGGCTGAGAAC-3′; RAB20 forward 5′-CTATGATGTGAATCACCGGCAG-3′ and RAB20 reverse 5′-GGTCCCCAGCGTCCATATTG-3′.

### 2.8. Cell Proliferation, Migration, Colony Formation, and Wound Healing Assays

Tetrazolium salt cell viability assay was conducted to explore the proliferation potential of RAB20 cells. A total of 2 × 10^3^ PSCC cells (shRAB20 and NC) were seeded into 96-well plates in 100 μL DMEM, incubated with 10 μL Cell Counting Kit-8 (CCK-8, Dojindo, Kumamoto, Japan) solution for 2 h, and counted by a microplate reader (optical density: 450 nm) for 7 consecutive days. For the cell migration assay, 10^5^ PSCC cells were seeded in the upper compartment of 24-well transwell chambers in serum-free DMEM, with 10% FBS-DMEM in the lower compartment. The number of migrating cells was counted after a 24 h incubation. For the colony formation assay, 2000 cells were cultured in 6-well plates for 14 days, and the cell colonies were counted by ImageJ (National Institutes of Health, Bethesda, MD, USA). For the wound healing assay, when PSCC cells covered the 6-well plates, cross lines were drawn with 1000 μL pipette tips. After incubation with serum-free DMEM for 16 h, the wound-healing capacity was determined by measuring the size of the gaps.

### 2.9. Cell Cycle Assay

For cell cycle profiling, Penl2 and 149rca PSCC cells (shRAB20 and shNC groups) were synchronized for 24 h in serum-free DMEM, followed by fixation in 70% ethanol overnight. Then, tumor cells (10^5^ per tube) were stained with the Cell Cycle Staining Kit (KeyGEN, Nanjing, China) according to the manufacturer’s instructions. Flow cytometry (ACEA NovoCyte™, San Diego, CA, USA) was used to explore the percentage of cells in each cell cycle phase and the data were analyzed by NovoExpress™ (Walnut Creek, CA, USA). Cell cycle distribution is presented as histograms from three independent experiments.

### 2.10. Xenograft Assay

Five- to seven-week-old male BALB/c nude mice (Jiangsu GemPharmatech Co., Ltd., Nanjing, China) were randomly divided into the RAB20-sh3 and NC groups (*n* = 7) and housed under the same conditions. Each mouse was injected subcutaneously with tumor cells (10^6^ Penl2 cells transfected with RAB20-sh3 or shNC in 150 μL phosphate buffer saline) and sacrificed three weeks later. Tumors were harvested for further IHC and WB assays. In addition, mice were euthanized when weight loss over 15% or a tumor size exceeding 1500 mm^3^ occurred. The Experimental Animal Ethics Committee of SYSUCC (L102022021001D, approved on 12 July 2021) approved the animal protocols.

## 3. Results

### 3.1. The Expression Profiles of mRNAs in PSCC Tissues

To investigate the different profiles of mRNA expression as PSCC progresses, we compared eight pairs of PSCC tissues (including adjacent normal (N), primary carcinoma (PCA) and metastatic lymph node tissues (LM) by using a comprehensive transcriptome microarray (Appendix A). The expression levels of 19 mRNAs were found to be upregulated in tumor tissues, especially with the highest expression in the LM group, as shown in (Figure 1A and Appendix A). Subsequently, the corresponding 19 genes were verified by an external GEO database (GSE57955), which included 39 penile cancer tissues and five corresponding normal tissues [21]. The results indicated that AIM2, MM9, and RAB20 (log_2_ fold changes: 4.24, 3.94, and 2.01, respectively) were the top three genes with a significantly high expression in PSCC tissues (Figure 1B and Appendix A). Our previous study has demonstrated that AIM2 was an oncogene associated with poor survival in PSCC [24]. MMP9 was upregulated in most of the tumors and facilitated cell invasion and metastasis with clear molecular mechanisms [25]. However, the clinical significance and biological functions of RAB20 remained blank in PSCC and were further explored in this study.

### 3.2. RAB20 Is Overexpressed in PSCC Cell Lines and Tissues

To evaluate the expression level of RAB20 in PSCC cells and tissues, WB and IHC were performed to detect the expression of RAB20 protein. Compared with that in normal epithelial cells or tissues, RAB20 protein was overexpressed in five PSCC cell lines and tumor tissues (Figure 1C,D). The IHC results indicated that RAB20 was abundant in the cytoplasm but sporadic staining in the nucleoplasm (Figure 1E and Appendix A). RAB20 was faintly or slightly stained in the normal squamous epithelium. Conversely, RAB20 was diffusely and strongly stained in corresponding PSCC cells (Figure 1E and Appendix A).

Next, the qRT-PCR assay was performed to detect the mRNA levels of RAB20 in a larger cohort of 78 PSCC tissues and 21 normal tissues. Compared with normal samples, RAB20 mRNA levels were significantly elevated in PSCC tissues (*t* = 3.023, *p* = 0.009) (Figure 1F). Especially, 15 pairs of corresponding samples were included in the cohort, and the same results were observed that RAB20 mRNA levels were significantly higher in tumor tissues than they were in corresponding normal tissues (*t* = 3.779, *p* < 0.001) (Figure 1G).

### 3.3. Overexpression of RAB20 Was Associated with Poor Clinical Features in PSCC

To identify the association between RAB20 expression and clinical significance, 259 paraffin-embedded PSCC sections were subjected to IHC assays. As detected by IHC, RAB20 had weak and faint expression in the normal cells, while it was strongly and diffusely expressed in tumor cell cytoplasm with sporadic staining in the nucleoplasm. The staining patterns are shown in Figure 1H. The IHC scoring criteria are described in the section Materials and Methods, and the frequency in each subgroup is shown in Appendix A. In this cohort, a total of 100 patients (38.6%) died from PSCC, with a median follow-up time of 81.0 months (IQR: 46.0–134.0). The IHC results showed that 147/259 (56.8%) PSCC patients had high RAB20 expression, while 112 (43.2%) patients had low RAB20 expression (Table 1). Chi-square analysis demonstrated that overexpression of RAB20 was positively associated with increased pT and pN status, metastasis, extranodal extension (ENE), and clinical stage (all *p* < 0.01), indicating poor PSCC patient prognosis (Table 1).

Then, survival analyses showed that RAB20 overexpression led to poor CSS outcomes (*p* < 0.001) (Figure 2A), and consistent results were found in the pT2-pT4, pN+, ENE, and pathological grade subgroups (all *p* < 0.05) (Figure 2B–F). Other clinical features, including pT2-pT4 subgroup, pN status, pathological grade subgroup, ENE were also related to CSS (all *p* < 0.05, Appendix A). Furthermore, multivariate analysis demonstrated that RAB20 expression (*p* = 0.011, HR = 2.090; 95% Cl: 1.183–4.692) is an independent prognostic factor for poor PSCC patient CSS outcomes (Table 2). These findings suggest that the overexpression of RAB20 is a novel marker associated with poor PSCC clinical features.

### 3.4. RAB20 Knockdown Inhibits Cell Proliferation and Cell Cycle Progression in PSCC

To investigate the oncogenic function of RAB20, we first suppressed the expression of RAB20 by short hairpin RNAs (shRNAs) in PSCC cell lines (Penl2 and 149rca), and the knockdown efficiency was measured by WB (Figure 3A). CCK-8 cell proliferation and colony formation assays demonstrated that RAB20-silenced Penl2 and 149rca cells demonstrated dramatically impaired cell growth compared with that of negative controls (Figure 3B,C). Knockdown of RAB20 inhibited the migration of PSCC cells, as shown in (Figure 3D). Additionally, in vivo experiments demonstrated that knockdown of the expression of RAB20 in Penl2-sh3 cells resulted in smaller and lighter tumors in BALB/c nude mice (Figure 3E). Moreover, GSEA was utilized to dissect the expression difference of mRNA-seq between RAB20-sh3 Penl2 cells and negative control cells. The results showed significant enrichment in the cell replication and cell cycle checkpoint pathways, indicating a prominent role for RAB20 in cell proliferation (Figure 3F).

### 3.5. RAB20 Overexpression Promotes Cell Proliferation in PSCC

To further validate the proliferative capability of RAB20, RAB20 overexpression plasmids were transfected into RAB20-sh3 and wild-type PSCC cells (Figure 4A). Cell proliferation and colony formation assays indicated that overexpression of RAB20 promoted cell replication and increased the number of clones in vitro. Meanwhile, rescuing the expression of RAB20 in RAB20-sh3 PSCC cells relieved the inhibition of cell proliferation and stimulated cell growth (Figure 4B,C).

### 3.6. Knockdown of RAB20 Induced G2/M Cell Arrest in PSCC

Flow cytometry assays were performed to further investigate the role of RAB20 in the cell cycle phase. The results showed that the proportion of cells in the G2 phase was dramatically increased from 10.67% to 26.18% in Penl2 cells and from 16.15% to 27.30% in 149rca cells when RAB20 was inhibited (Figure 5A). WB verified that the expression of the critical G2/M regulatory proteins cyclinB1 and cdc2 [26] was significantly downregulated after the knockdown of RAB20 in Penl2 and 149rca cells. In contrast, the expression levels of G1/S cell cycle regulators, such as cyclin D1 and the CDK2-cyclin E1 complex [26], were not significantly altered (Figure 5B). These findings indicate that RAB20 silencing induces G2/M cell arrest in PSCC cells, which contributes to the resulting inhibition of cell proliferation.

### 3.7. RAB20 Regulates the Cell Cycle via the Chk1/cdc25c/cdc2-cyclinB1 Pathway in PSCC

Recently reported evidence suggests that the Chk1/cdc25c-dependent and Chk2/p53-dependent molecular pathways contribute to the G2/M transition [27,28,29]. Therefore, we first measured cdc25c and p53 phosphorylation levels by WB. The results revealed that knockdown of RAB20 in PSCC cells increased the expression of Chk1, which activated the phosphorylation of cdc25c at Ser216 and then reduced the expression of non-phosphorylated cdc25c [28,30]. However, the phosphorylation levels of p53 and its downstream protein p21 were not affected (Figure 5C,D). Previous studies demonstrated that non-phosphorylated cdc25c (activated form) was responsible for dephosphorylating p-cdc2 and activated cdc2 to combine with cyclinB1 protein, triggering mitosis [30,31,32]. Consistent with these findings, we found that the reduction of cdc25c in Penl2-shRAB20 cells decreased the expression of cdc2 and cyclinB1 inhibiting the formation of cdc2-cyclinB1 complex (Figure 5C). Moreover, the expression of cdc2 and cyclinB1 in RAB20-silenced tumors of xenograft nude mice were decreased dramatically compared with the shNC group (Appendix A). Therefore, we detected that RAB20 depended on Chk1/cdc25c/cdc2-cyclinB1 pathway to induce G2/M cell cycle arrest.

To further validate the molecular mechanism of RAB20-mediated G2/M cell cycle arrest, we performed rescue experiments by repressing Chk1 expression in shRAB20 PSCC cells. Figure 5E shows that the percentage of shRAB20 Penl2 and 149rca cells transfected with Chk1-siRNA in the G2 phase decreased from 23.08% to 7.63% and 24.74% to 5.42%, respectively (Figure 5E), indicating that RAB20-mediated G2/M cell cycle arrest was Chk1/cdc25c-dependent. With the inhibition of Chk1 in shRAB20 PSCC cells, the phosphorylation level of cdc25c (non-activated form) dropped significantly, whereas the expression of cdc25c elevated and then promoted the expression of the cdc2-cyclinB1 complex (Figure 5F). Taken together, our results indicated that RAB20 induced cell cycle arrest at the G2/M phase, which depended on Chk1 mediating the cdc25c/cdc2-cyclinB1 pathway in PSCC cells.

## 4. Discussion

PSCC is more common in developing countries, such as in China, than in the United States and Europe. Chinese PSCC patients account for one-third of new cases worldwide, and PSCC is a substantial health concern due to its mental and physical effects [33,34]. Although the TNM staging system, especially the pT stage, is a convenient and effective clinical prognostic tool for evaluating survival, the ability to perform a precise and individualized assessment with this approach is limited by tumor heterogeneity [7,8,35]. Recently, several sequencing studies in Europe and the Americas have revealed the clinically relevant genomic alterations in PSCC and provided a better understanding of tumor progression and targets for therapy [7,8,9,10,11]. However, potential prognostic biomarkers and specific mechanisms of action remain unknown due to the lack of experimental data and large-cohort clinical validation; in particular, analysis of the genomic expression pattern in Chinese PSCC patients had not been previously reported [7,8,9,10,11]. Therefore, for the first time, we performed whole-transcriptome microarray profiling of paired PSCC tissues to explore the genomic landscape in Chinese patients and to investigate the potential biomarkers and mechanisms of PSCC.

In this study, we found that the expression of RAB20, a small GTPase family member located on chromosome 13q34 [22], was upregulated in PSCC matched tumor tissues, especially in metastatic lymph nodes. qPCR, WB, and IHC further confirmed that RAB20 was overexpressed in the cytoplasm of five PSCC cell lines and 78 PSCC tumor tissues compared with that in the corresponding normal controls. The results highlight that RAB20 might be a crucial oncogene participating in the development and progression of PSCC, although the roles of RAB20 in PSCC have not yet been reported.

Recent studies have revealed that RAB20, which plays a role in the control of endocytotic vesicle transport, is involved in the progression of multiple cancers [15,16,17,36]. Amillet et al. [15] first identified that the overexpression of RAB20 in pancreatic intraductal neoplasia lesions is an early event in the course of pancreatic cancer progression [15]. Habermann et al. [17] demonstrated that RAB20 overexpression and amplification indicated genomic instability in colorectal adenomas and was correlated with high-grade histopathological features and tumor recurrence [17]. In addition, the overexpression of RAB20 in triple-negative breast cancer has been associated with advanced disease stages and poor patient prognosis [16].

To further explore the clinical significance of RAB20 expression in PSCC, correlation and survival analyses were conducted on 259 PSCC patients, the largest cohort reported to date, with a median follow-up time of over six years. The overexpression of RAB20 in PSCC was positively associated with advanced clinicopathological features and a shorter 5-year CSS time. Moreover, RAB20 was found to be a strong independent prognostic indicator of poor clinical outcomes, identifying PSCC patients with an increased risk for tumor progression and a shorter survival time. These findings underscore the clinical significance of RAB20 in PSCC and imply that RAB20 plays oncogenic roles in tumor progression.

A recent study on neuronal network formation showed that RAB20, as a novel regulator, participated in neurite outgrowth and cell proliferation [37]. Liu et al. [36] found that the restoration of RAB20 expression in hepatocellular carcinoma cells inhibited cell growth, motility, and metastasis. To further investigate the oncogenic functions of RAB20 in regulating the malignant phenotype, validation and rescue experiments were conducted in vitro and in vivo. We observed that knockdown of RAB20 in our newly established PSCC cell lines repressed colony formation, cell proliferation, and migration along with repressing tumor growth in xenograft nude mice. Interestingly, GSEA showed that RAB20-silenced Penl2 cells were not only enriched in cell proliferation pathways but also in cell cycle checkpoint pathways. Habermann et al. [17] reported that RAB20 amplification triggered EGFR recycling and promoted cell proliferation by increasing the formation of cyclin A-CDK2 complexes in the S/G2 cell cycle phases [17]. Therefore, we explored the cell cycle distribution by flow cytometry, and the results indicated that RAB20-silenced PSCC cells exhibited G2/M phase cell cycle arrest but unaffected G1/S transition. Similarly, knockdown of RAB20 did not inhibit the expression of the CDK2-cyclinE1 complex, the key effector in the G1/S phase, but significantly suppressed cdc2-cyclinB1 levels at the G2/M transition [26]. Overall, we found that RAB20-mediated cancer progression could be tumor-type specific and promotes PSCC proliferation by regulating G2/M cell cycle checkpoints.

Cdc25c phosphatase promotes the mitotic cell G2/M transition by triggering cdc2 dephosphorylation to activate the cdc2-cyclinB1 complex [30,32]. The activation of cdc25c requires phosphorylation within the N-terminal domain at Thr48, Thr67, Ser122, Thr130, or Ser216 sites, which is regulated by the checkpoint protein kinases Chk1 and Chk2 and p53 pathways [32]. We found that the knockdown of RAB20 did not alter the phosphorylation of p53 or the transcriptional activation of p21 (a downstream protein in the p53 pathway) [27]. However, in shRAB20 PSCC cells, the Chk1-mediated phosphorylation of cdc25c at Ser216 was increased, and the cdc2-cyclinB1 complex was inhibited. Subsequently, we measured the cell cycle stage distribution and protein expression when Chk1 was repressed in shRAB20 PSCC cells. We found a remarkable decrease in the proportion of cells in G2, and the expression levels of the cdc2-cyclinB1 complex and p-cdc25c (Ser216) were restored. The results further demonstrate that RAB20 induces G2/M phase cell arrest via the Chk1/cdc25c pathway in PSCC.

Our findings suggest that the overexpression of RAB20 in PSCC may be an initiator of the cellular dysregulation, which acts as an upstream regulator in the G2/M cell cycle phase promoting tumor proliferation and progression. However, the detailed mechanisms by which RAB20 regulates Chk1 still remain unclear and need to be further investigated. In addition, we found that the 5-year CSS rate of PSCC patients with low RAB20 expression was 81.9% (Table 2), but 57.1% (12/21) of them died within 30 months due to the rapid progression of tumors. These patients may be attributed to the tumor heterogeneity and the aberrant expression of driver genes, such as CDKN2A mutation or EGFR amplification. Other limitations also deserve to mention. We focused on the oncogenes that were upregulated in the comprehensive genomic sequencing and did not study the potential tumor suppressor genes that might promote the progression of PSCC. Moreover, more efforts at multiple centers are required to determine the potential value of RAB20 as a biomarker for PSCC.

## 5. Conclusions

In the current study, we explored the potent oncogene RAB20, which is overexpressed in PSCC and is associated with advanced clinicopathological features and poor patient prognoses. High RAB20 expression promotes tumor progression and cell proliferation, inducing G2/M cell cycle arrest via the Chk1/cdc25c/cdc2-cyclinB1 pathway. RAB20 could be a potential therapeutic target and serve as a novel prognostic indicator of PSCC patient outcomes.

## Figures and Tables

**Figure 1 cancers-14-01106-f001:**
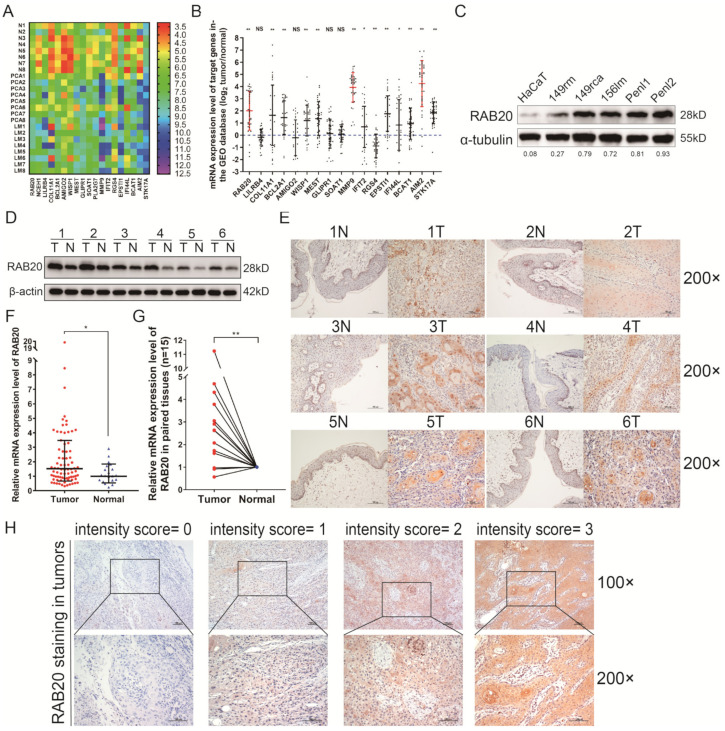
Expression levels of RAB20 in PSCC tissues and cell lines. (**A**) The comprehensive CGP analysis of eight pN+ PSCC patients indicated 108 co-upregulated genes in the PCA group and LM group, of which 19 genes were more highly expressed in the LM group. The heatmap shows the expression pattern of the 19 target genes. (**B**) Among the 19 genes in the GSE57955 database, AIM2, MMP9 and RAB20 were the top 3 genes with a significantly high expression level in PSCC tissues (log_2_ tumor/control: 4.24, 3.94 and 2.01 respectively), and followed with STK17A, COL11A1, BCL2A1, MEST and WISP1 (FC: 1.86, 1.64, 1.45, 1.37 and 1.20 respectively). IFIT2, IFI44L, RGS4 and BCAT1 were downregulated in the GSE57955 database (FC: 0.68, 0.84, 0.88 and 0.95 respectively). (**C**,**D**) The RAB20 protein was overexpressed in PSCC cell lines and tumor tissues in pairs. (**E**) IHC staining indicated that RAB20 was highly expressed in tumors compared with the corresponding normal tissues (magnification: 200×). (**F**,**G**) The mRNA levels of RAB20 were upregulated in 78 tumor tissues compared with those in 21 normal tissues (15 pairs). GAPDH was selected as the internal reference gene, and the results are presented as the means ± SDs of three independent experiments (**H**) RAB20 staining scores were multiplied by the staining intensity and the staining area. The standard staining intensity score of RAB20 in the cytoplasm was 0 for no staining, 1 for weak staining, 2 for clear staining and 3 for strong staining. * *p* < 0.05, ** *p* < 0.01. FC, fold change; HaCaT, human immortalized keratinocytes; IHC, immunohistochemistry; NS, not significant. The uncropped blots are shown in Appendix A.

**Figure 2 cancers-14-01106-f002:**
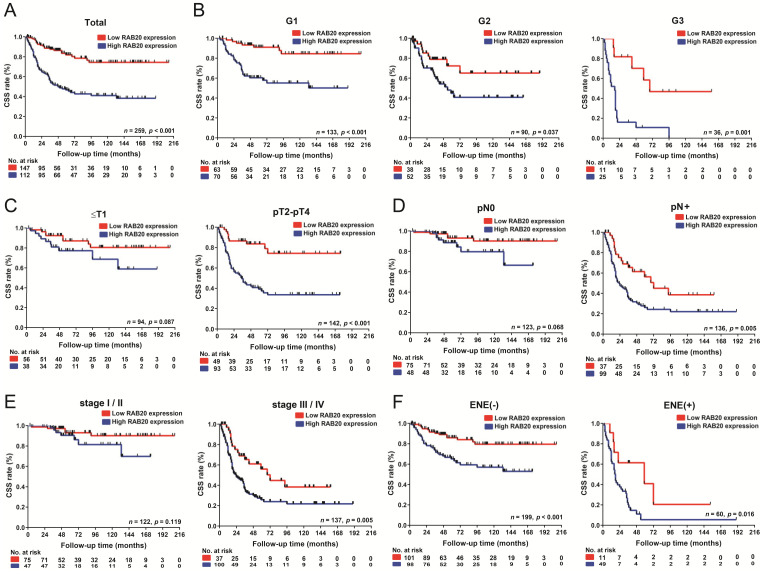
Survival analysis between RAB20 expression and clinical features in 259 PSCC patients. Kaplan–Meier survival analysis was performed to determine the RAB20 expression level in PSCC patients. (**A**) High RAB20 expression indicated a significantly lower CSS rate in the entire cohort, (**B**) pathological grade subgroup, (**C**) pT2-pT4 subgroup, (**D**) positive lymph node metastasis subgroup, (**E**) clinical stage III/IV subgroup, and (**F**) ENE subgroup of PSCC patients (*p* < 0.05). CSS, cancer-specific survival; ENE, extranodal extension.

**Figure 3 cancers-14-01106-f003:**
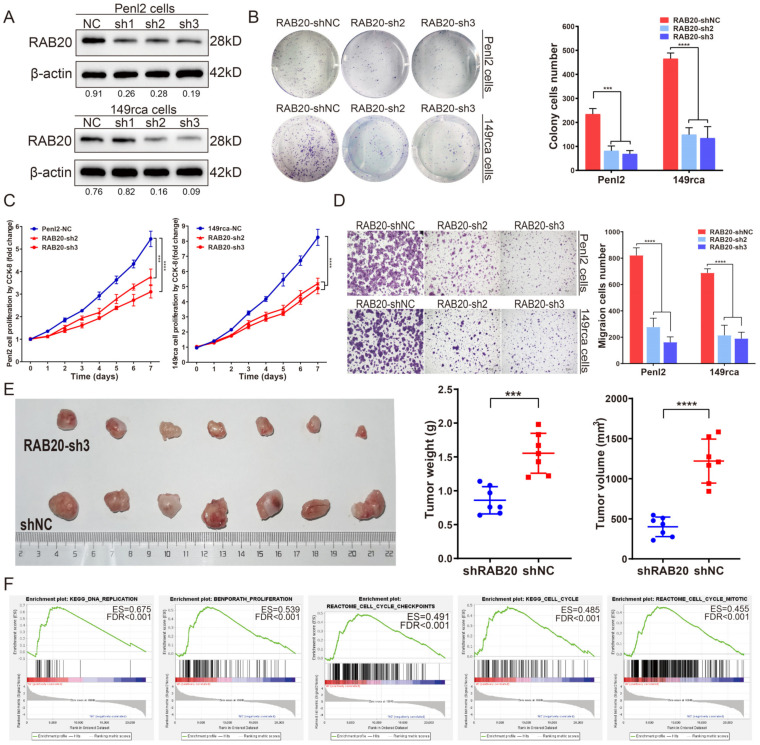
Knockdown of RAB20 inhibited the proliferation of PSCC cells in vivo and in vivo. (**A**) Western blotting was performed to examine the knockdown efficacy in Penl2 and 149rca RAB20-silenced cells. (**B**) Knockdown of RAB20 in PSCC cells inhibited cell colony formation, (**C**) cell proliferation, (**D**) cell migration in vitro, (**E**), and tumor growth in vivo. (**F**) GSEA showed that knockdown of RAB20 in Penl2 cells significantly influenced the expression of proteins in the cell cycle and proliferation. Statistics are presented as the means ± SDs of three independent experiments. * *p* < 0.05, ** *p* < 0.01, *** *p* < 0.001, **** *p* < 0.0001. GSEA, gene set enrichment analysis; IHC, immunohistochemistry; NC, negative control. The uncropped blots are shown in Appendix A.

**Figure 4 cancers-14-01106-f004:**
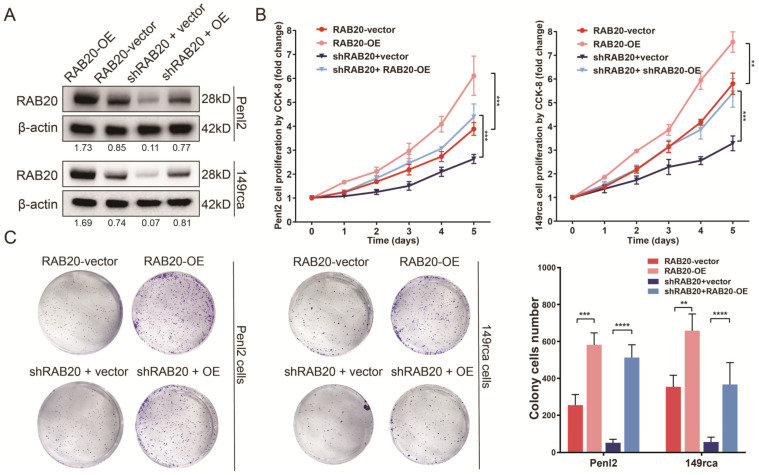
Overexpression of RAB20 promoted and rescued the proliferative ability of PSCC cells. (**A**) Western blotting was performed to detect the overexpression efficiency of RAB20 in wild type or transfection with RAB20-sh3 PSCC cells. (**B**,**C**) CCK-8 proliferation assays and colony formation assays revealed that RAB20 overexpression promoted cell proliferation and colony formation in Penl2 and 149rca cells and relieved the proliferative capability of cells restrained by RAB20 knockdown. Statistics are presented as the means ± SDs of three independent experiments. ** *p* < 0.01, *** *p* < 0.001, **** *p* < 0.0001. OE, overexpression. The uncropped blots are shown in Appendix A.

**Figure 5 cancers-14-01106-f005:**
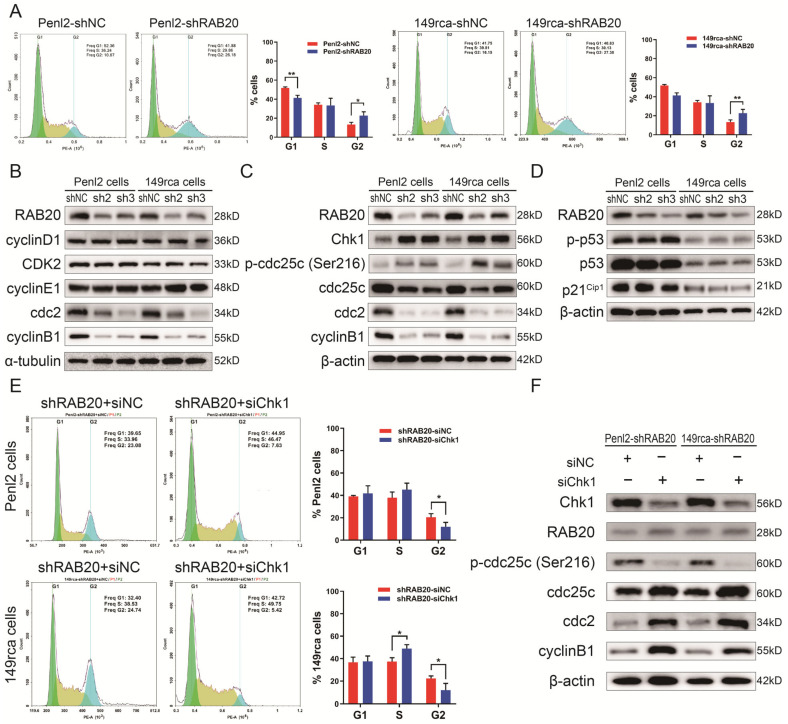
Knockdown of RAB20 induced G2/M cell arrest via the Chk1/cdc25c/cdc2-cyclinB1 pathways. (**A**) Penl2 and 149rca cells transfecting with RAB20-sh3 increased the proportion of cells in the G2 phase and inhibited cell proliferation. (**B**) Knockdown of RAB20 significantly reduced the expression of the G2/M checkpoint proteins cdc2 and cyclinB1 but did not influence the G1/S phase. (**C**,**D**) Knockdown of RAB20 induced cell cycle arrest at the G2/M phase via the Chk1/cdc25c/cdc2-cyclinB1 pathway rather than via the p53 pathway. (**E**,**F**) Knockdown of Chk1 dramatically attenuated G2/M cell arrest in RAB20-sh3 cells and recovered the expression of the G2/M checkpoint proteins cdc2 and cyclinB1. Flow cytometry was performed in triplicate and the results are presented as the mean ± SD. * *p* < 0.05, ** *p* < 0.01. The uncropped blots are shown in Appendix A.

**Table 1 cancers-14-01106-t001:** Association of RAB20 expression with clinicopathological features of 259 PSCC patients.

		RAB20 IHC Staining			
Variable	PSCC Cohort (*n* = 259), %	Low Expression (*n* = 112), %	High Expression (*n* = 147), %	χ^2^	*p*-Value ^a^
Age				0.424	0.515
<55	149 (57.5)	67 (25.9)	82 (31.7)		
≥55	110 (42.5)	45 (17.4)	65 (25.1)		
pT status				17.137	0.002 ^b^
≤pT1 ^c^	94 (36.3)	56 (21.6)	38 (14.7)		
pT2	35 (13.5)	13 (5.0)	22 (8.5)		
pT3	96 (37.1)	34 (13.1)	62 (23.9)		
pT4	11 (4.2)	2 (0.8)	9 (3.5)		
Tx	23 (8.9)	7 (2.7)	16 (6.2)		
pN status				35.484	0.000
N0	123 (47.5)	75 (29.0)	48 (18.5)		
N1	32 (12.4)	11 (4.2)	21 (8.1)		
N2	32 (12.4)	13 (5.0)	19 (7.3)		
N3	72 (27.8)	13 (5.0)	59 (22.8)		
Metastasis				8.679	0.003 ^b^
M0	244 (94.2)	111 (42.9)	133 (51.4)		
M1	15 (5.8)	1 (0.4)	14 (5.4)		
Clinical stage ^d^				39.060	0.000
Stage I	65 (25.1)	46 (17.8)	19 (7.3)		
Stage II	57 (22.0)	29 (11.2)	28 (10.8)		
Stage III	56 (21.6)	20 (7.7)	36 (13.9)		
Stage IV	81 (31.3)	17 (6.6)	64 (24.7)		
Histology				3.322	0.190
G1	133 (51.4)	63 (24.3)	70 (27.0)		
G2	90 (34.7)	38 (14.7)	52 (20.1)		
G3	36 (13.9)	11 (4.2)	25 (9.7)		
ENE				19.743	0.000
No	199 (76.8)	101 (39.0)	98 (37.8)		
Yes	60 (23.2)	11 (4.2)	49 (18.9)		

^a^ Chi-square test; ^b^ Fisher’s exact test; ^c^ Included Ta, Tis and pT1 patients; ^d^ Clinical stage was based on the AJCC Cancer Staging Manual and TNM Staging System for Penile Cancer (8th ed., 2017); ENE, extranodal extension; PSCC, penile squamous cell carcinoma.

**Table 2 cancers-14-01106-t002:** Univariate and multivariate analyses of clinical and pathological features in 259 PSCC patients.

		Univariate Analysis ^a^			Multivariate Analysis ^b^	
Variable	Total *n*	Events (%)	5-Year CSS Rate (95% Cl)	*p*-Value	Hazard Ratio (95% Cl)	*p*-Value
Age				0.037		0.271
<55	149	51 (34.2)	0.665 (0.583–0.747)		Reference	
≥55	110	49 (44.5)	0.542 (0.442–0.642)		1.304 (0.813–2.091)	
pT status ^c^				0.000		0.039
≤pT1	94	18 (19.1)	0.829 (0.747–0.911)	Reference	Reference	-
pT2	35	15 (42.9)	0.539 (0.365–0.713)	0.001	2.135 (1.027–4.438)	0.042
pT3	96	41 (42.7)	0.589 (0.483–0.695)	0.000	1.970 (1.101–3.526)	0.022
pT4	11	10 (90.9)	0.000	0.000	3.307 (1.290–8.479)	0.013
Histology				0.000		0.109
G1	133	37 (27.8)	0.746 (0.670–0.822)	Reference	Reference	-
G2	90	35 (38.9)	0.550 (0.428–0.672)	0.010	1.005 (0.567–1.781)	0.986
G3	36	28 (77.8)	0.248 (0.097–0.399)	0.000	1.766 (0.936–3.332)	0.079
pN status				0.000		0.000
N0	123	13 (10.6)	0.910 (0.853–0.967)	Reference	Reference	-
N1	32	11 (34.4)	0.664 (0.492–0.836)	0.000	2.135 (1.027–4.438)	0.005
N2	32	16 (50.0)	0.481 (0.289–0.673)	0.000	1.970 (1.101–3.526)	0.000
N3	72	60 (83.3)	0.112 (0.020–0.204)	0.000	3.307 (1.290–8.479)	0.000
Metastasis				0.000		0.008
M0	244	85 (34.8)	0.655 (0.590–0.720)		Reference	
M1	15	15 (100)	0.000		2.686 (1.291–5.588)	
Clinical stage ^d^				0.000		
Stage I	65	7 (10.8)	0.900 (0.816–0.984)	Reference	Excluded ^e^	
Stage II	57	5 (8.8)	0.942 (0.879–1.000)	0.776		
Stage III	56	21 (37.5)	0.636 (0.499–0.773)	0.000		
Stage IV	81	67 (82.7)	0.104 (0.020–0.188)	0.000		
ENE				0.000		0.109
No	199	52 (26.1)	0.754 (0.689–0.819)		Reference	
Yes	60	48 (80.0)	0.116 (0.008–0.224)		1.293 (0.600–2.786)	
RAB20				0.000		0.011
Low expression	112	21 (18.7)	0.819 (0.739–0.899)		Reference	
High expression	147	79 (53.7)	0.467 (0.383–0.551)		2.090 (1.183–4.692)	

^a^ Log-rank test; ^b^ Cox regression model (Tx patients excluded, *n* = 236); ^c^ 23 Tx patients were excluded and stratified analysis found there was no significant difference between pT2/pT3 (χ^2^ = 0.005; *p* = 0.944); ^d^ Stratified analysis revealed no significant difference between stage I/stage II (χ^2^ = 0.081; *p* = 0.776); ^e^ Clincial stage was excluded from the Cox regression model as it was represented by the TNM stage. CSS, cancer-specific survival.

## Data Availability

The authenticity of this article has been validated by uploading the key raw data onto the Research Data Deposit platform (www.researchdata.org.cn), with the approval RDD number as RDDB2021956079.

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
