# Peer review of "RAB20 Promotes Proliferation via G2/M Phase through the Chk1/cdc25c/cdc2-cyclinB1 Pathway in Penile Squamous Cell Carcinoma"

_cancers, 2022, doi:10.3390/cancers14051106_

Round 1

Reviewer 1 Report

The current study investigated the expression and oncogenic effect of RAB20 PSCC. Clinical samples, multiple in vitro cell lines and in vivo animal model were used. Results suggested RAB20 promotes PSCC progression, predicting advanced disease with poor outcomes of PSCC. This study is interesting and topic is relevant. Also, experiments were well conducted and results were detailed. However, I have several comments to their work.

  1. Besides knockdown experiments, it is better to add overexpression model to confirm RAB20’s function.
  2. For animal experiments, is it possible to investigate the expression of RAB20 and its downstream targets?

Author Response

Comment 1: Besides knockdown experiments, it is better to add overexpression model to confirm RAB20’s function.

Reply 1:

We greatly appreciated the valuable advice and guidance from the reviewer to help us optimize the study. Followed with the reviewer’s good suggestion, RAB20 plasmids were transfected into Penl2 and 149rca cells to construct the RAB20 overexpression cell lines (revised Figure 4A). Cell proliferation and cell colony formation assay were used to measure the ability of cell growth, and the results showed that overexpression of RAB20 increased the ability of cell growth and proliferation (revised Figure 4B-C). Also, rescue experiment was performed in Penl2-RAB20sh3 and 149rca-RAB20sh3 cells by transfecting with RAB20 plasmids. As expected, overexpression of RAB20 in shRAB20 cells rescued the ability of RAB20 promoting cell growth and proliferation (revised Figure 4A-C). Thank you for the excellent suggestion!

Changes in the text:

  1. Materials and Methods”: The information of the RAB20 plasmid were added in 2.3. Cell lines, culture conditions and transfection methods.
  2. “Results”: The results of RAB20 overexpression and rescued experiments were add in 3.5. RAB20 overexpression promotes cell proliferation in PSCC.

Comment 2: For animal experiments, is it possible to investigate the expression of RAB20 and its downstream targets?

Reply 2: Thanks for this good suggestion. Because the tumor tissues were frozen to store in -80℃ fridges, it caused an inconvenience of performing IHC. Thus, we isolated the protein extracts of frozen tumor tissues and detected the protein expressions of RAB20 and its downstream molecules (revised Figure S4). Consisted with the results in PSCC cell lines, cyclinB1 and cdc2 proteins (the critical complex of G2/M cell cycle checkpoint) were significantly downregulated in the shRAB20 group.

Changes in the text:

  1. “Results”: The results of the expression of RAB20 and its downstream targets were described in 7. RAB20 regulates the cell cycle via the Chk1/cdc25c/cdc2-cyclinB1 pathway in PSCC.
  2. Figure S4 was added in the Supplementary.

Reviewer 2 Report

Thank you for your nice paper. A few questions to ask

  1. You mentioned that 8 pairs of pN+ PSCC matched tissues 83 were retrieved for comprehensive genomic profiling (CGP). Ninety-nine fresh frozen samples (78 tumor tissues and 21 normal tissues) were collected for mRNA and protein extraction. Did you process the entire 99 samples and were RAB20 overexpressed in all samples?
  2. Paraffin-embedded tumor sections from 259 PSCC patients were re-analyzed by two independent pathologists (CCB and LLL), and immunohistochemistry (IHC) staining was performed. How do you determine the quality of samples kept over the years? How many are penile tissues vs nodal tissues?
  3.  What is the purpose of xenograft assay?
  4. Of the 259 PSCC patients especially those with poor CSS, how many have under-expression or no expression of RAB20? If so, how do you explain this observation?
  5. Where do you propose RAB20 to sit in the overall genome sequence of penile cancer tumorigenesis? Given that RAB20 regulates Chk1 in G2/M cell cycle arrest, is this observation mean RAB20 is an initiator of the cellular dysregulation or perhaps antagonist to some potential tumor suppressor genes? 

Author Response

Comment 1: You mentioned that 8 pairs of pN+ PSCC matched tissues 83 were retrieved for comprehensive genomic profiling (CGP). Ninety-nine fresh frozen samples (78 tumor tissues and 21 normal tissues) were collected for mRNA and protein extraction. Did you process the entire 99 samples and were RAB20 overexpressed in all samples?

Reply 1: Thank you for this question! Yes, we processed all 99 (78 tumor and 21 normal tissues) samples for PCR analysis. Our result showed that the average level of RAB20 mRNA (n=78) was higher in tumor tissues than it did in normal tissues (n=21, as shown in Figure 1F). It didn’t mean that RAB20 mRNA was overexpressed in every tumor sample or RAB20 mRNA was downexpressed in every normal sample. Among them, 15 tumor and 15 corresponding normal samples were paired tissues. The same results were observed in the 15 paired tissues (Figure 1G). Thanks again!

Changes in the text:

  1. “Results”: The description of results were revised more clearly in 2. RAB20 is overexpressed in PSCC cell lines and tissues.

Comment 2: Paraffin-embedded tumor sections from 259 PSCC patients were re-analyzed by two independent pathologists (CCB and LLL), and immunohistochemistry (IHC) staining was performed. How do you determine the quality of samples kept over the years? How many are penile tissues vs nodal tissues?

Reply 2: Thanks for this question! We will first retrieve the HE sections of primary tumor of the PSCC patients. After pathologists re-confirming the diagnosis of PSCC, the corresponding wax blocks were selected, sliced to 4 µm tissue sections and followed by RAB20 staining. Two independent pathologists (CCB and LLL) performed quality control and calculated RAB20 IHC scores independently. The unqualified slides will be re-sectioned. In our cohort, we only retrieved the sections of primary PSCC tumors, and did not evaluate the was blocks of lymph node metastases.

Changes in the text: None.

Comment 3: What is the purpose of xenograft assay?

Reply 3: Thanks for this question! Xenograft assay was used to assess the role of RAB20 in PSCC cell proliferation in vivo. Although there is no orthotopic penile cancer model, we subcutaneously injected human penile cancer cells into nude mice to establish the xenograft assay.

Changes in the text: None

Comment 4: Of the 259 PSCC patients especially those with poor CSS, how many have under-expression or no expression of RAB20? If so, how do you explain this observation?

Reply 4: Thanks for this question! As described in the text, 112 (43.2%) patients had low RAB20 expression. The 5-year CSS rate of PSCC patients with low RAB20 expression was 81.9% (Figure 2A and Table 2). A total of 21 patients died and 57.1% (12/21) of them died within 30 months due to the progression of tumors. Among them, one PSCC patient was absence of RAB20 staining.

It is recognized internationally that cancer progression is a multi-factorial, multi-step and multi-stage process, which involves a variety of changes in gene expression. RAB20 is one of the radical oncogene in the tumor development and progression of PSCC, but is not unique. These patients may be attributed to the tumor heterogeneity and the aberrant expression of other driver genes, such as CDKN2A mutation or EGFR amplification. Besides, the clinical and pathological factors such as the present of lymph node metastasis (N stage) and pathological stage (G stage) were strong predictors in prognosis.

Changes in the text:

1.Discussion: We have explained the reasons and added in the last paragraph of the discussion.

Comment 5: Where do you propose RAB20 to sit in the overall genome sequence of penile cancer tumorigenesis? Given that RAB20 regulates Chk1 in G2/M cell cycle arrest, is this observation mean RAB20 is an initiator of the cellular dysregulation or perhaps antagonist to some potential tumor suppressor genes?

Reply 5: Thank you for the inspiring question. Our results finally suggested an upstream role of RAB20 in the overall genome sequence of penile cancer tumorigenesis. Our result just revealed that RAB20 had a regulatory role in cell cycle arrest by regulating Chk1-cdc25c-cdc2/cyclinB1 pathway. It might indicated that RAB20 could be an initiator of the cellular dysregulation, and regulated Chk1 expression or activation. However, the underlying mechanism of how RAB20 directly or indirectly inhibits the activation of Chk1 remains unknown, and has to been investigated in our further study. Our result couldn’t make a conclusion whether RAB20 is an antagonist to some potential tumor suppressor genes. But the inspiring question deserves more exploratory efforts to answer! Overall, we regard that RAB20 is an upstream oncogene in PSCC regulating the G2/M cell cycle phase. Thanks again!

Changes in the text:

1.Discussion: We have explained it and added as limitation in the last paragraph of the discussion.

Reviewer 3 Report

Journal

Cancers (ISSN 2072-6694)

Manuscript ID

cancers-1519289

Type

Article

Title

RAB20 promotes proliferation via G2/M phase through the Chk1/cdc25c/cdc2-cyclinB1 pathway in penile squamous cell carcinoma

The authors present finding RAB20 as an overexpressed oncogene in penile squamous cell carcinoma (PSCC). Expression by IHC was found to correlated independently with survival using a large cohort of PSCC patients, and RAB20 depletion resulted in G2/M arrest dependent on Chk1/cdc25c/cdc2-cyclinB.  The paper provides new insight into drivers of aggressive behavior and outcome in PSCC and provides a potential biomarker.

The major concern relates to how the IHC scoring was calculated and how that correlates with staining shown in the manuscript. Understanding this is critical for the biomarker determination.

The authors describe high IHC staining of RAB20 as tumor cytoplasmic staining score greater than 4, but figure 1H that shows staining patterns indicates only intensity score of up to 3. How does this intensity score correlate with the cytoplasmic staining score. And related, in Fig S1 scoring of IHC is not well explained and is critical for the paper as to how the cut point for high vs. low RAB20 was determined using the software.

Axis for figure S1B should be labeled on x axis with every integer and on y axis every 20

Given the differences in carcinogenesis based on HPV status, it would be interesting for the authors to correlate HPV status with outcome and with RAB20 expression.

Additional concerns that should be addressed:

Fig 1a and b. The authors should clearly state which of the 19 genes were and were not significantly altered in tumors.

Fig 1d. The authors should quantify the bands then normalize to the reference band to determine the fold change in protein across the tumors

Fig 1e. Please indicate on the figure the magnification for each picture and show higher power of the tumors and normal as well as describe the staining both cytoplasmic and nuclear with examples in normal and tumor and justify reason for choosing cytoplasmic staining only if there is nuclear staining

Fig 1f and g. The authors should state that mRNA expression was determined by RTPCR relative to GAPDH, if this is what is represented.

Fig 1h. High IHC staining described as tumor cytoplasmic staining score greater than 4, but figure 1H staining patterns shows only intensity score of up to 3. How does the intensity score correlate with the staining score.

Fig S1.  Scoring of IHC is not well explained and is critical for the paper as to how the cut point for high vs. low RAB20 was determined using the software.

Fig S1B. Axis should be better labeled - x axis with every integer and y axis every 20

Additional notes:

In the manuscript, the authors should describe the CCK8 assay as tetrazolium salt cell viability assay and in methods the brand name and company should be indicated.

Fig 2 if there is a PSCC line with low RAB20, it would be interesting to see if overexpression increased proliferation or colony formation

Fig 3a should quantify the KD by WB

Fig 3F Does GSEA for 149rca match that shown for penl2 cells

Line 196 AIM2 was an oncogene with poor survival in PSCC – should be associated with poor survival

Line 237 and 239 rather than both p<0.05 - should be all p<0.05

Author Response

Comment 1: High IHC staining described as tumor cytoplasmic staining score greater than 4, but figure 1H staining patterns shows only intensity score of up to 3. How does the intensity score correlate with the staining score? Scoring of IHC is not well explained and is critical for the paper as to how the cut point for high vs. low RAB20 was determined using the software.

Reply 1: Thank you for bringing this error to our attention! The description is incorrect. In fact. The IHC staining scores of RAB20 was multiplied by the staining intensity and the staining area. Staining intensity scores were based on the RAB20 cytoplasm staining (Figure 1H, 0 for no staining, 1 for weak staining, 2 for moderate staining and 3 for strong staining) and staining area defined as 1 for 1-10%, 2 for 11-40%, 3 for 41-70%, and 4 for 71% above. The frequency of each groups were listed in Figure S2A-B. The cutoff value was calculated by X-Tile software (Version 3.6.1) by standard Monte Carlo simulation methods [1]. In our cohorts, the cut-off value of RAB20 IHC scores was 4 points (Figure S2). It indicated that 0-4 points were regarded as low RAB20 expression and 6-12 points were high RAB20 expression.

[1] Camp R L, Dolled-Filhart M, Rimm D L. X-tile:a new bio-informatics tool for biomarker assessment and outcome-based cut-pointoptimization [J]. Clin Cancer Res, 2004, 10(21): 7252-9.

Changes in the text:

  1. Materials and Methods” and “Results”: The criteria of RAB20 IHC staining scores were revised in 5. Immunohistochemistry assay and 3.3. Overexpression of RAB20 was associated with poor clinical features in PSCC.
  2. Figure S2 was revised. We increased the frequency of staining intensity, staining area and total score of RAB20 respectively.

Comment 2: The authors should clearly state which of the 19 genes were and were not significantly altered in tumors (Fig 1A and B). The authors should quantify the bands then normalize to the reference band to determine the fold change in protein across the tumors. The authors should state that mRNA expression was determined by RTPCR relative to GAPDH, if this is what is represented (Fig 1F and G).

Reply 2: Thanks for the suggestion. The description of significantly upregulated and downregulated genes were added in the legend of Figure 1, and the raw resulted were added in Table S2 and Table S3. We have quantified the bands in the knockdown and overexpression efficiency of cell lines (Figure 1C, Figure 3A, Figure 4A and Figure S4). We have stated that RAB20 mRNA expression was determined by RT-qPCR relative to GAPDH in the legend of Figure 1 and Methods.

Changes in the text:

  1. Tables and Figures: Table S2 and Table S3 were added, and We have quantified the bands in the knockdown and overexpression efficiency of cell lines (Figure 1C, Figure 3A, Figure 4A and Figure S4).
  2. Materials and Methods: We stated that the mRNA expression levels were normalized against GAPDH in Figure legend of Fig 1, and the Methods part (2.7. Quantitative real-time polymerase chain reaction assay).

Comment 3: Please indicate on the figure the magnification for each picture and show higher power of the tumors and normal as well as describe the staining both cytoplasmic and nuclear with examples in normal and tumor and justify reason for choosing cytoplasmic staining only if there is nuclear staining (Fig 1E)?

Reply 3: Thank you for the suggestion. We replaced the higher power (magnification: 200X) and clearer RAB20 IHC staining images of paired PSCC tissues (Figure 1E), and the original images were shown in Figure S1 (magnification: 100X) for a large landscape. RAB20 had weak and faint expression in the squamous epithelial cells, while it was strongly and diffusely expressed in tumor cell cytoplasm with sporadic staining in the nucleoplasm (Figure 1E). RAB20 was an vesicle trafficking related protein, localized in the Golgi apparatus and vesicles in the cytoplasm. We found that nucleoplasmic RAB20-positive tumor cells were also strongly positively expressed in the cytoplasm. The reason of RAB20 positive expression in the nucleoplasm might be the non-specific staining. Therefore, we chose cytoplasmic staining as the scoring criterion.

Changes in the text:

  1. “Results”: The description of RAB20 in normal and tumor cells were added in3. Overexpression of RAB20 was associated with poor clinical features in PSCC.

Comment 4: Axis should be better labeled - x axis with every integer and y axis every 20 (Fig S1B).

Reply 4: Figure S1D was generated automatically by the X-tile software. In order to describe the RAB20 IHC staining scores clearly, we have added the distribution and frequency of raw data with the pie graph (revised Figure S2A-B).

Changes in the text:

1.Supplementary Tables and Figures: We have revised Figure S2.

Comment 5 (Additional notes): In the manuscript, the authors should describe the CCK8 assay as tetrazolium salt cell viability assay and in methods the brand name and company should be indicated.

Reply 5: We have revised the corresponding parts of the manuscript.

Changes in the text:

1.Materials and Methods: The description of CCK8 assay was revised in the 2.8. Cell proliferation, migration, colony formation and wound healing assays.

Comment 6 (Additional notes):Fig 2 if there is a PSCC line with low RAB20, it would be interesting to see if overexpression increased proliferation or colony formation

Reply 6: We greatly appreciated the valuable advice and guidance from the reviewer to help us optimize the study. Followed with the reviewer’s good suggestion, RAB20 plasmids were transfected into Penl2 and 149rca cells to construct the RAB20 overexpression cell lines (revised Figure 4A). Cell proliferation and cell colony formation assay were used to measure the ability of cell growth, and the results showed that overexpression of RAB20 increased the ability of cell growth and proliferation (revised Figure 4B-C). Also, rescue experiment was performed in Penl2-RAB20sh3 and 149rca-RAB20sh3 cells by transfecting with RAB20 plasmids. As expected, overexpression of RAB20 in shRAB20 cells rescued the ability of RAB20 promoting cell growth and proliferation (revised Figure 4A-C). Thank you for the excellent suggestion!

Changes in the text:

  1. Materials and Methods”: The information of the RAB20 plasmid were added in 2.3. Cell lines, culture conditions and transfection methods.
  2. “Results”: The results of RAB20 overexpression and rescued experiments were add in 3.5. RAB20 overexpression promotes cell proliferation in PSCC.

Comment 7 (Additional notes): Fig 3a should quantify the KD by WB

Reply 7: We quantified the protein expression of target genes and added the sign of KD in the figures.

Changes in the text: Figure 1C-D, Figure 3A, Figure 4A, Figure 5 and Figure S4 were revised.

Comment 8 (Additional notes): Fig 3F Does GSEA for 149rca match that shown for penl2 cells

Reply 8: Thank for the suggestion. The GSEA analysis was based on differential gene expression levels between RAB20-shNC and RAB20-sh3 cell lines. However, RNA-seq was only performed between Penl2-shNC cells and Penl2-RAB20sh3 cells. Although we are unable to perform verification of GSEA in 149rca cells directly, we verified the Penl2 and 149rca cell lines separately by WB in the subsequent experiments of G2/M phrase cell cycle regulation.

Changes in the text: None.

Comment 9 (Additional notes): Line 196 AIM2 was an oncogene with poor survival in PSCC, should be associated with poor survival. Line 237 and 239 rather than both p<0.05 - should be all p<0.05.

Reply 9: We have revised the corresponding parts of the manuscript.

Round 2

Reviewer 2 Report

Thank you for addressing my concerns. 

Reviewer 3 Report

additional data has clarified points of concern.  Acceptable for publication.